# Synthesis of Extracellular L-lysine-α-oxidase along with Degrading Enzymes by *Trichoderma* cf. *aureoviride* Rifai VKM F-4268D: Role in Biocontrol and Systemic Plant Resistance

**DOI:** 10.3390/jof10050323

**Published:** 2024-04-28

**Authors:** Anna Yu. Arinbasarova, Alexander S. Botin, Alexander G. Medentsev, Kirill V. Makrushin, Alexandre A. Vetcher, Yaroslav M. Stanishevskiy

**Affiliations:** 1G.K. Skryabin Institute of Biochemistry and Physiology of Microorganisms, Russian Academy of Sciences, 142290 Pushchino, Russia; medentsev-ag@rambler.ru (A.G.M.); makr82@rambler.ru (K.V.M.); 2Institute of Biochemical Technology and Nanotechnology (IBTN), Peoples’ Friendship University of Russia na. P.Lumumba (RUDN), 6 Miklukho-Maklaya St., 117198 Moscow, Russia; botin_as@pfur.ru (A.S.B.); stanishevskiy-yam@rudn.ru (Y.M.S.); 3N.V. Sklifosovsky Institute of Emergency Medicine, 129090 Moscow, Russia

**Keywords:** *Trichoderma*, degradative enzymes, cellulase, chitinase, protease, glucanases, L-lysine-α-oxidase, hydrogen peroxide, pipecolinic acid, antimicrobial activity, biocontrol, systemic plant resistance

## Abstract

When cultivating on wheat bran or deactivated fungal mycelium as a model of “natural growth”, the ability of *Trichoderma* to synthesize extracellular L-lysine-α-oxidase (LysO) simultaneously with cell-wall-degrading enzymes (proteases, xylanase, glucanases, chitinases, etc.), responsible for mycoparasitism, was shown. LysO, in turn, causes the formation of H_2_O_2_ and pipecolic acid. These compounds are known to be signaling molecules and play an important role in the induction and development of systemic acquired resistance in plants. Antagonistic effects of LysO have been demonstrated against phytopathogenic fungi and Gram-positive or Gram-negative bacteria with dose-dependent cell death. The antimicrobial effect of LysO decreased in the presence of catalase. The generating intracellular ROS in the presence of LysO was also shown in both bacteria and fungi, which led to a decrease in viable cells. These results suggest that the antimicrobial activity of LysO is due to two factors: the formation of exogenous hydrogen peroxide as a product of the enzymatic oxidative deamination of L-lysine and the direct interaction of LysO with the cell wall of the micro-organisms. Thus, LysO on its own enhances the potential of the producer in the environment; namely, the enzyme complements the strategy of the fungus in biocontrol and indirectly participates in inducing SAR and regulating the relationship between pathogens and plants.

## 1. Introduction

Fungi of the genus *Trichoderma* play an important role in the formation of soil microbiocenoses because they produce many metabolites, including lytic enzymes (cellulases, chitinases, glucanases, and proteases) that degrade the cell walls of the fungi and bacteria, as well as antibiotics of various chemical structures [1,2,3].

It is also known that fungi are capable of synthesizing exogenous L-amino acid oxidases, including L-lysine-α-oxidase (LysO) [4,5,6,7]. LysO attracts the interest of researchers due to perspectives for its applications in biotechnology and medicine [8,9].

Some kinetic properties of the enzyme such as the high enzyme activity, “narrow” substrate specificity, and high selectivity, as well as absolute L-stereospecificity, represent a good possibility for biotechnological applications in protein engineering [10,11].

Multiple in vitro and in vivo investigations have revealed the great potential of LysO for tumor enzyme therapy [12,13,14]. Therefore, LysO from *Trichoderma* cf. *aureoviride* Rifai VKM F-4268D showed a dose-dependent cytotoxic effect in vitro towards different cancer cell lines: erythromyeloblastic leukemia K562, breast cancer MCF7, and colon cancer LS174T. The antitumor effects of LysO in vivo were shown by tumors of different species: breast adenocarcinoma Ca755, ascitic hepatoma 22, melanoma B16, colon carcinoma AKATOL, and cervical cancer RSHM5 [14].

It is known that *Trichoderma* fungi grow using various media containing simple carbohydrates (glucose, sucrose, starch, etc.), as well as complex substrates, including wheat bran [6,7,15,16] or the cell wall of phytopatogenic fungi [3,5,17]. To grow on these complex substrates, fungi need to synthesize a complex of lytic enzymes (xylanases, glucanases, proteases, etc.) that degrade natural polymers into energy-rich monomers that can subsequently be metabolized. The ability to synthesize such cell-wall-degrading enzymes causes *Trichoderma* species to be highly interactive in nature. 

The synthesis of LysO by *Trichoderma* fungi was not considered to be in an appropriate relationship with these complexes and remains beyond the scope of studying the role of the enzyme in the natural environment. This issue is also important from the position of the development of biotechnological processes for the production of extracellular LysO as a promising medicine.

The purpose of this work is to analyze some aspects of the synthesis of L-lysine-α-oxidase simultaneously with cell-wall-degrading enzymes by *Trichoderma* cf. *aureoviride* Rifai VKM F-4268D and to reveal the functional role of the enzyme in the participation of the producer in biocontrol and inducing systemic plant resistance.

## 2. Materials and Methods

### 2.1. Micro-Organism

The object of research was the fungus *Trichoderma* cf. *aureoviride* Rifai VKM F-4268D from the All-Russian Collection of Micro-organisms (VKM, Institute of Biochemistry and Physiology of Microorganisms, Russian Academy of Sciences, Pushchino, Russia) capable of synthesizing extracellular LysO.

### 2.2. Growth of Micro-Organism

The fungus *Trichoderma* was grown in Czapek’s medium containing (g/L) KH_2_PO_4_, 1.0; MgSO_4_∙7H_2_O, 0.5; KCl, 0.5; and yeast autolysate, 0.05. Various sources of carbon were used such as wheat bran (3%), xylan (2%), chitosan (2%), or glucose (2%). Deactivated mycelium of the *Fusarium decemcellulare* VKM F-1169 was also used in a concentration of 3% (dry weight). For this, the grown mycelium was inactivated by heating to 100 °C, followed by separation from the growth medium by centrifugation and drying. Cultivation was performed in 750 mL flasks containing 100 mL of the medium for 10–13 days in a shaker (220 rpm) at 29 °C.

### 2.3. Enzyme Biosynthesis Evaluation

The efficiency of enzyme biosynthesis was estimated by measuring relevant activity accumulation in the growth medium expressed in international units (U/mL).

To control the accumulation of metabolites in the growth medium, samples were taken at certain intervals and centrifuged at 6000× *g* for 20 min, and enzyme activity was determined in the supernatant. LysO activity was assayed at 22 °C by the rate of hydrogen peroxide production in 20 mM Tris-phosphate buffer (pH 7.8) in the presence of o-dianisidine (0.2 mM), peroxidase (5 µg/mL), and L-lysine (2 mM) with a Shimadzu spectrophotometer (E 436 = 8.3 mM ^−1^ cm^−1^). The enzyme quantity catalyzing the oxidation of 1 µmol lysine per minute was taken as a unit of activity. Proteolytic activity in the samples was assayed with stained casein [18].

The activities of chitinases, cellulases, b-1,3-glucanases, b-1,6-glucanases, and xylanases were determined according to the technique for measuring the amount of reducing sugar [13], with some subsequent modifications. The reaction mixture consisted of 1.5 mL of 50 mM potassium phosphate buffer (pH 6.7) and substrate (0.5%)—chitin, laminarin, pustulan, or xylan. After 0.1 mL of culture medium was added, mixture was incubated at 50 °C for 60 min, followed by a drop of 0.1 mL of dinitrosalicylic acid reagent (Sigma, St. Louis, MO, USA) and incubation at 90–100 °C for 20 min. When the color developed, OD_540_ was fixed.

The extinction coefficients of glucose, xylose, and N-acetyl-glucosamine were determined. An enzymatic unit of β-1,3-glucanase, β-1,6-glucanase, was defined as the amount of enzyme that catalyzes the release of reducing sugar groups that are equivalent to 1 µmol of glucose per min. One unit of xylanase activity was defined as the amount of enzyme that releases 1 µmol of reducing sugar equivalents to xylose per min. One unit of chitinase activity was defined as the amount of enzyme that can release 1 µmol of reducing sugar equivalents to N-acetylglucosamine per h.

### 2.4. Isolation and Purification of LysO

L-lysine-alpha-oxidase was isolated from the fungus *Trichoderma* cf. *aureoviride* Rifai VKM F-4268D using the technique described earlier [19]. Homogenous enzyme preparation of high purity (310-fold) with high specific activity (equal to 90 U/mg of protein) was used in the present work.

### 2.5. Hydrogen Peroxide Determination

Hydrogen peroxide in the culture liquid was assayed by the change in o-dianisidine absorption in the presence of peroxidase in a Shimadzu spectrophotometer. The reaction mixture containing o-dianisidine (0.2 mM) in 25 mM Tris-phosphate buffer (pH 7.5) was added to an aliquot of the culture liquid (0.1–0.2 mL) up to 1 mL. The reaction was started by adding peroxidase (up to 0.5 µg/mL). Upon color development, the sample was centrifuged for 5 min at 12,000× *g*.

### 2.6. Determination of Pipecolic Acid

Trichloroacetic acid was added to an aliquot of the *Trichoderma* growth medium (to a concentration of 5%), and then the mixture was centrifuged for 10 min at 10,000× *g*.

The precipitate was discarded, and metabolites were extracted from the supernatant with ethyl acetate (1:10, *v*:*v*). Then, the extract was evaporated, and the residue was dissolved in methanol and analyzed using mass spectrometry.

Mass spectra of the compounds were recorded on an LCQ Advantage MAX quadrupole mass spectrometer (Thermo Finnigan, Bremen, Germany), using a single-channel syringe pump for direct injection of the sample into the chamber for chemical ionization at atmospheric pressure. Detailed information about the structure of compounds was obtained by analyzing MS/MS spectra at the collision energy of 20–40%. Mass spectrometric analysis was carried out by Dr. Baskunov B.P. (IBFM RAS).

### 2.7. Antimicrobial Activity

Antimicrobial activity was studied by the requirements of the State Pharmacopoeia of the Russian Federation or the Clinical and Laboratory Standards Institute’s guidelines [20]. The test micro-organisms were obtained from All-Russian Collection of Micro-organisms (VKM, IBPM RAS) or FSBI Tarasevich State Research Institute of Standardization and Control of Biological Medicines (GISK).

Next, micro-organisms were used: Gram-positive bacterium—*Bacillus subtilis* VKM B-720, *Enterococcus durans* VKM B-603, and *Staphylococcus epidermidis* GISK 33; Gram-negative—*Escherichia coli* K12, *Pseudomonas aureofaciens* VKM B-1249, and *Pseudomonas aeruginosa* GISK 453; and fungi—*Fusarium decemcellulare* VKM F-1179, *Rhizoctonia solani* VKM F-895, and *Aspergillus niger* VKM F-1119.

Bacteria were cultured in Luria–Bertani (LB) broth containing (g/L) 10 g tryptone, 5 g yeast extract, and 6.4 g KCl (Sigma Aldrich, St. Louis, MO, USA). Fungi were grown in the medium containing (g/L) 15.0 malt extract, 12.75 maltose, 0.78 peptone, and 2.75 dextrin, with pH 4.0.

The suspension of the test micro-organism (1 mL) (1 × 10^−5^ CFU mL^−1^) was placed into test tubes with 10 mL of corresponding liquid nutrient medium, and then 0.2 mL (~30 U) of LysOx was added. In the control variants, 0.2 mL of the appropriate nutrient medium was added to the test tubes instead of the enzyme. Growth of micro-organisms was monitoring at a temperature of 29–32 °C and shaking at 160 rpm for 3–5 days. The growth of micro-organism was assessed visually. If the growth of test micro-organism took place in the control, and it was not observed in the presence of the enzyme, this indicated the antimicrobial effect of LysO against test micro-organism.

To determine antimicrobial activity, agar nutrient media were also used. Sterile disks from filter paper were wetted with sterile solutions of LysO (~1 U), L-lysine (10 µM), or catalase (10 U) and dried in a sterile box. Then, Petri dishes with agar medium were plated (lawn inoculation) with the test microbial cultures. The inoculated agar plates were overlayed with the dry filter disks containing LysO, L-lysine, or catalase and exposed at 32 °C for 1–3 days. The antimicrobial effect appeared as a transparent zone of varying widths around the paper disk.

All assays were performed in triplicate including the growth and sterility control.

### 2.8. Measurement of Intracellular Reactive Oxygen Species (ROS)

The formation of reactive oxygen species (ROS) in cells was studied with the fluorescent dye dihydro-2′,7′-dichlorofluorescein diacetate using a Hitachi MPF-4 spectrofluorimeter (Tokyo, Japan) (λ_excit_ = 485 nm, λ_emis_ = 528 nm) [15]. Liquid nutrient medium 5/5 was inoculated with cells of either the bacterium *S. aureus* ATCC 6538 or the phytopathogenic fungus *R. solani* VKM F-895 and incubated at 30 °C. Growth was monitored by measuring the optical density of the bacterial culture every 3 h and the fungal culture every 12 h. When the optical density of the cultures reached 0.5, LysOx was added in the concentration of 5 or 10 µg/mL.

The percentage of viable cells in the culture samples was determined using the diagnostic kits LIVE/DEAD BacLight Bacterial Viability Kit and LIVE/DEAD Kit for Yeast and Fungi (Thermo Fisher Scientific, Waltham, MA, USA). The respective dyes were mixed in equal volumes (10 µL), and 3 µL of the resultant mixture was added to the analyzed culture samples. Then, the samples were incubated at room temperature in the dark for 30 min and assayed using a Zeiss Axio Lab.A1 fluorescent microscope (Oberkochen, Germany). Viable and dead cells in the field of view of the microscope were counted in quintuplicate for each sample.

### 2.9. Statistics

Statistical analyses were performed by the results of the Shapiro–Wilk test of a normal distribution. Data were further analyzed using a one-way ANOVA, followed by post hoc two-sided Dunnett’s tests. The results reflect the mean (±standard deviations (SD)) of three independent experiments.

## 3. Results

Table 1 and Figure 1A,B represent the results of measuring the activities of extracellular enzymes of *Trichoderma* growing on Czapek’s medium containing various sources of carbon. During growth on wheat bran or xylan or chitosan, the fungus appeared to synthesize Lys-Ox and varied degradative enzymes—xylanases, glucanases, proteases, and chitinases—simultaneously. Despite chitin not being a component of wheat bran, when it was used as a carbon source, the fungus has synthesized both LysO and degradative enzymes. On the other hand, chitinase has been synthesized during growth not only on chitosan, but also on xylan or wheat bran as the substrates.

In addition, in the presence of glucose, the synthesis of the enzymes under consideration was not observed—neither LysO nor degradative enzymes. This is a key point in the regulation of protein synthesis and the triggering of “protective” mechanisms, apparently associated with nutritional stress when consuming complex substrates, such as wheat bran [15].

And, to complete, all the metabolites above have also been synthesized if the fungus was growing in medium containing deactivated mycelium of *Fusarium decemcellulare* VKM *F*-1179 (Table 1).

In Figure 1B, one can see the dynamics of the accumulation of LysO and proteolytic activity, which hydrolyze the protein components of wheat bran and ensure the presence of amino acids in the growth medium, including lysine. Other than lysine, more than 10 amino acids, including lysine, were detected in the medium [7].

The process of LysO biosynthesis was also accompanied by the formation of hydrogen peroxide—a product of the enzymatic oxidative deamination of L-lysine. The appearance of this product use coincides with the starting of LysO biosynthesis (on days 2–3 of fungus growth) (Figure 1).

In addition to lytic enzymes, LysO, and H_2_O_2_, pipecolic acid was found in the growth medium containing wheat bran or mycelia deactivated using the mass spectrometry technique (Figure 2).

It was noted that the appearance of pipecolic acid coincides in time with the starting of the biosynthesis of LysO and the disappearance of lysine.

Pipecolic acid is the product of the cyclization of α-keto-ε-aminocaproic acid, that, in its turn, is the product of lysine oxidation by LysO. But α-keto-ε-aminocaproic acid was not detected in the growth media.

Thus, extracellular LysO and degradative enzymes, including protease, hydrogen peroxide, and pipecolic acid, were found in the environment of the fungus during growth on wheat bran or deactivated mycelia. They may be just those metabolites that are involved in ensuring the survival and competition of the fungus under natural conditions.

### Antimicrobial Activity

To characterize the relevant functional role of the producer under natural conditions, the antimicrobial effects of LysO have been demonstrated.

LysO turned out to inhibit the growth of all the micro-organisms tested, including Gram-positive (*Bacillus subtilis* VKM B-720, *Enterococcus durans* VKM B-603, and *Staphylococcus epidermidis* GISK 33) and Gram-negative bacteria (*Escherichia coli* K12, *Pseudomonas aeruginosa* GISK 453, and *Pseudomonas aureofaciens* VKM B-1249). The inhibition of fungi germination was also observed *(Fusarium decemcellulare* VKM F-1179, *Rhizoctonia solani* VKM F-895, and *Aspergillus niger* VKM F-1119).

In Figure 3, the inhibitory effect of LysO on the growth of *B. subtilis* VKM B-720 (A) and *E. durans* VKM B-603 (B) on agar nutrient media is presented as an example. And, also, as an example, the inhibiting growth of fungus *R. solani* VKM F-895 is demonstrated in Figure 4. It can be seen in Figure 3 that zones of lyses were formed, indicating cell death, which increased with an increasing enzyme dose. Similar dependences on the enzyme concentration were observed for all micro-organisms under consideration, including fungi and bacteria.

To clarify the question of the possible mechanism of action of LysO on micro-organisms, the antimicrobial effect of the enzyme in the presence and the absence of catalase was demonstrated when test micro-organisms were cultivating on an agar nutrient medium (Figure 5).

It can be seen that the size of the lytic zone caused by LysOx on the lawn of the catalase-negative bacterium *E. durans* VKM B-603 increased by 4 times after lysine was added (Figure 5A (1 and 2)). The subsequent addition of catalase (Figure 5A (3)) led to a diminishment in the size of the transparent zone by 2.5 times (reducing growth inhibition). The same results were obtained with another catalase-negative bacterium *P. aureofaciens* (Figure 5B).

The effects of catalase and/or lysine on the antagonistic activity of LysO against catalase-positive bacterium (*B. subtilis*, *S. epidermidis*, or *E. coli*) were weakly noticeable or absent entirely (not shown in the pictures).

It should be emphasized that, in all the cases, the addition of catalase did not lead to the removal of the antimicrobial effect of LysOx completely.

A further examination of the mechanism of antimicrobial activity of LysOx consisted of measuring the intracellular content of ROS. One can see (in Figure 6A,B) that, in the presence of LysOx, the fluorescence of dihydro-2′,7′-dichlorofluorescein diacetate increases, indicative of the enhanced formation of intracellular ROS. In this case, cell death occurred—when incubated with LysO at a concentration of 10 μg/mL, the proportion of dead *S. epidermidis* cells reached 90% after 18 h, and *R. solani* after 48 h.

## 4. Discussion

When cultivating on wheat bran or deactivated fungal mycelium as a model variant of “natural growth,” the ability of *Trichoderma* to synthesize cell-wall-degrading enzymes (proteases, xylanase, glucanases, chitinases, etc.) was established. This lytic activity allows *Trichoderma* to actively reproduce and exhibit competitive resistance [21,22].

It turned out that, along with the lytic activity above, *Trichoderma* produces extracellular LysO capable of (selectively) oxidizing lysine to form H_2_O_2_ and pipecolic acid.

It was previously shown [23,24,25,26] that H_2_O_2_ (or other reactive oxygen species) and pipecolic acid play a central role in the induction and development of systemic acquired resistance (SAR) in plants. Pipecolic acid induces the synthesis of plant hormones—jasmone and salicylic acids, which are responsible for the plant’s resistance to necrotrophs and biotrophs. In turn, H_2_O_2_ (and other reactive oxygen species) is a key element in the activation of mitogen-activated proteinases, which are also involved in the immune response of plants under biotic and abiotic stresses.

In an evolutionary sense, systemic resilience is based on several strategies that vary significantly from species to species. It is unlikely that there is a single regulation key. The biosynthesis of LysO along with cell-wall-degrading enzymes is one of the mechanisms. Other L-amino acid oxidases may also be involved, at least through the formation of H_2_O_2_. But LysO, moreover, can produce pipecolic acid, an elicitor of SAR.

*Trichoderma* fungi colonize the root epidermis and outer cortical layers of plants, and the biosynthesis of LysO in its territorial accessibility enhances the adaptive potential of both the plant and the producer by participating in the relationship between the pathogen and the plant.

It was also shown (Table 1) that, in the presence of glucose, the synthesis of all the enzymes above was not observed. This is an important point for understanding the tactics of micro-organism survival and the induction of defensive mechanisms.

When glucose is exhausted (or absent), the intracellular level of cAMP decreases, followed by CREB protein dephosphorylation, its dissociation from the CRE element, and the activation of gene transcriptions [27,28]. It is the decrease in the level of cAMP that leads to the synthesis of a variety of enzymes along with other adaptive reactions. That is why the fungus *Trichoderma* simultaneously synthesized some proteins, including cell-wall-degrading enzymes and LysO in the absence of glucose, when, certainly, the level of the nucleotide was low, namely, during growth on xylan, chitin, or wheat bran.

Lytic extracellular enzymes—proteases, xylanases, chitinases, and glucanases—are known to play an important role in the degradation of fungal cell walls and, thus, provide biocontrol activity [1,21,22,26,29].

In order to characterize the probable functional role for the producer under natural conditions, the antagonistic effect of LysO was examined. LysO was shown to inhibit the growth of various micro-organisms, including Gram-positive and Gram-negative aerobic bacteria and phytopatogenic fungi, with dose-dependent cell death occurring. On another position, a suppressed effect has been observed both for catalase-negative and catalase-positive bacteria (Figure 4 and Figure 6).

Some authors believe that the antimicrobial activity of oxidases is due to the hydrogen peroxide (one of the causal agents of oxidative stress) produced by these enzymes during the oxidation of their substrates [30]. The same mechanism of antimicrobial activity was also shown for the Ɛ-oxidases of L-lysine from *Rheinheimera* sp. and *Marinomonas mediterranea.*

Alternative mechanisms of antimicrobial activity of amino acid oxidases can be associated with their direct interaction with the cell wall and the induction of programmed cell death [4,5,15]. In any case, disturbances in various functions of the cell wall may occur (processes of polarization, transport of various metabolites, etc.).

To elucidate which mechanism of action is valid for LysO, we examined the influence of catalase on the action of LysO against various test agar cultures (Figure 6). The antimicrobial effect of LysO appeared to decreased in the presence of catalase, which is quite clearly expressed using catalase-negative bacteria. It should be repeated that, in all cases, the addition of catalase did not remove the antimicrobial effect of LysO completely.

We have also obtained results demonstrating the increase in the intracellular content of ROS in the presence of LysOx, which occurred in both bacteria and fungi. The generation of ROS led to a decrease in the number of viable cells.

A similar effect of phenylalanine oxidase from *T. harzianum* on the intracellular concentration of ROS in *R. solani* [4,5] was interpreted as a demonstration of mitochondrial dysfunction; this triggers the mitochondria-mediated apoptosis pathway, including cytochrome c release and the activation of apoptosis factors, caspases 3 and 9, and DNA fragmentation.

In addition, the accumulation of ROS in bacterial and fungal cells can lead to the peroxide oxidation of lipids and violate gene expression, also resulting in the death of microbial cells [31].

Our results obtained suggest that the antimicrobial activity of LysO is due to two factors, namely, (1) the formation of exogenous hydrogen peroxide as a result of the enzymatic deamination of L-lysine, and (2) direct interaction with components of the microbial cell wall and the accumulation of intracellular ROS.

The validity of the second factor is confirmed by the results of an investigation of the substrate specificity and the kinetic characteristics of LysO [10]. It was shown that LysO is active not only with lysine as a substrate, but also with its derivatives, such as diaminopimelic acid, that is a component of the peptidoglycan layer of the bacterial cell wall and provides an enzyme-binding site. The inhibition of LysO by substrate analogs, such as alanine, glutamine, and leucine, can also provide the binding of the enzyme with the cell wall due to the formation of a non-productive complex.

Earlier, we have shown [32] that, under the same conditions when growing on wheat bran, *T.* cf. *aureoviride* Rifai BKM F-4268D is also capable of producing an antimicrobial peptide. Despite the seeming independence, the synthesis of the peptide, lytic enzymes, and LysO are the parts of the total fungal strategy of the fungus in biocontrol and occur simultaneously, most likely in accordance with a common signal.

The first amino acid oxidase with antibacterial activity was isolated as far ago as in 1970 from the antagonistic effects of the *Crotalus adamanteus* venom [33]. But the study of such enzymes remains important until now due to the diversity of their functions and the mechanisms of their antimicrobial activity. The efficient action of amino acid oxidases on both Gram-positive and Gram-negative bacteria shows their promise for the development of remedies against antibiotic-resistant bacteria.

The antagonistic action of LysO from *Trichoderma* on the bacteria *B. subtilis* and *P. aureofaciens* (Figure 4 and Figure 5) should be taken into account in view of the fact that some species of the genera *Bacillus* and *Pseudomonas* favorably influence plants and soils and are extensively used in agriculture to improve soils and enhance the immune system of plants and their resistance to diseases, which, in turn, increases crop yield.

## 5. Conclusions

The characteristics of LysO above on their own are indicative of the high potential of the producer in the environment. The antimicrobial properties revealed, together with the formation of pipecolic acid and H_2_O_2_, predetermine the functional role of extracellular LysO. On the one hand, the synthesis of LysO, along with other extracellular proteins—lytic enzymes and antimicrobial peptides [32]—compliments the strategy of the fungus in biocontrol and provides an adaptive advantage in the competition with other micro-organisms in the natural environment. New aspects of biocontrol (in particular, of mycoparasitism) have been revealed; namely, the extracellular enzyme LysO is directly involved in the death of the pathogen, inducing a process similar to apoptosis. On the other hand, LysO can indirectly participate in inducing SAR and regulating the relationship between pathogens and plants.

## Figures and Tables

**Figure 1 jof-10-00323-f001:**
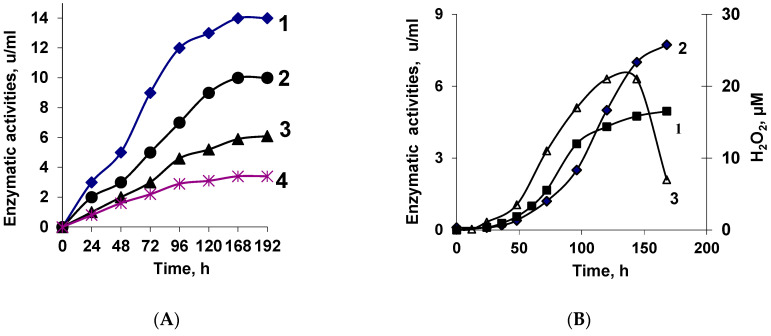
Enzyme activities (units/mL) of the extracellular proteins of *Trichoderma* cf. *aureoviride* Rifai BKM F-4268D grown on wheat bran. (**A**) 1—β-1,3-glucanases; 2—β-1,6-glucanases; 3—xylanases; and 4-chitinases. (**B**) 1—proteases; 2—L-lysine α-oxidase; and 3—H_2_O_2_. Since the confidential intervals do not exceed the size of the signs, they are not shown.

**Figure 2 jof-10-00323-f002:**
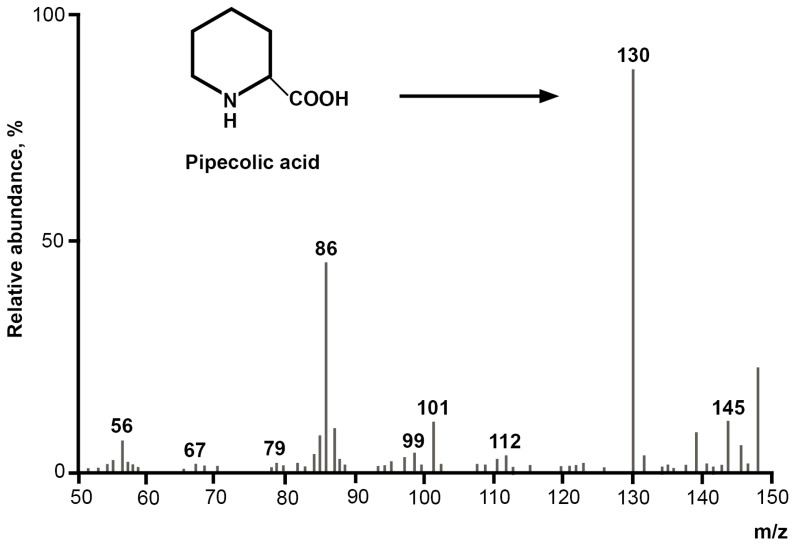
Mass spectrometric characterization of the metabolite in the growth medium of the fungus *Trichoderma* cf. *aureoviride* Rifai VKMF-4268D. Growth in submerged culture on wheat bran.

**Figure 3 jof-10-00323-f003:**
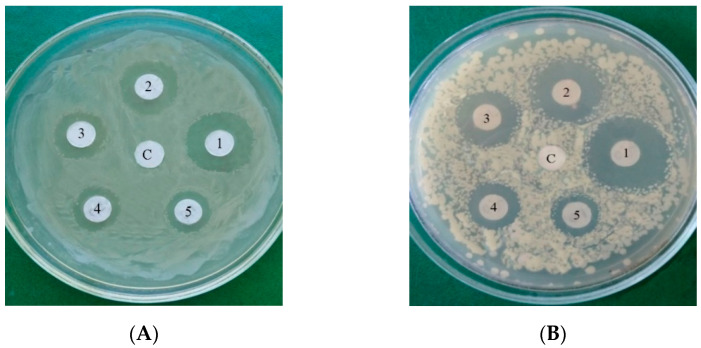
Inhibitory effect of LysOx on the growth of *B. subtilis* VKM B-720 (**A**) and of *E. durans* VKM B-603 (***B***). 1—5 U; 2—2.5 U; 3—1.25 U; 4—0.5 U; 5—0.25 U; C—control.

**Figure 4 jof-10-00323-f004:**
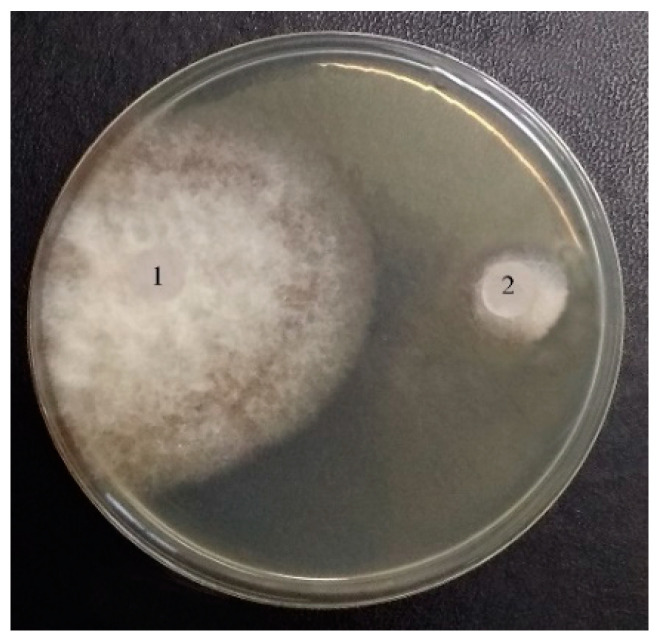
The influence of LysOx on the growth of *R. solani* VKM F-895. 1—without LysOx (control), 2—with LysOx (5 E).

**Figure 5 jof-10-00323-f005:**
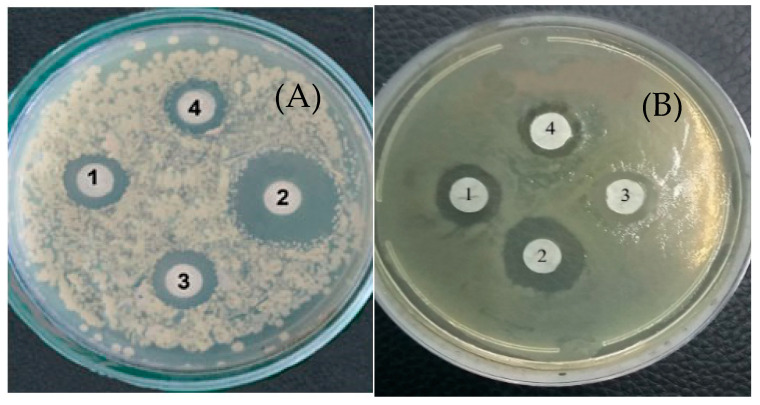
Influence of catalase on antimicrobial activity of LysO against catalase-negative bacteria. (**A**)—*E. durans* VKM B-603; (**B**)—*P. aureofaciens* VKM B-1249. 1—LysO (1 U); 2—LysO (1 U) + L-lysine (10 μM); 3—LysO (1 U) + L-lysine (10 μM) + catalase (10 μg); 4—LysO (1 U) + catalase (10 μg).

**Figure 6 jof-10-00323-f006:**
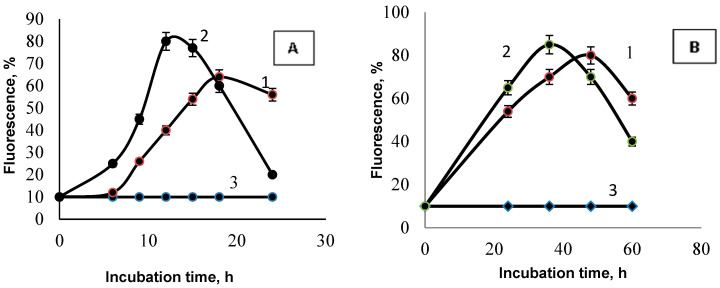
ROS generation in the cells *S. epidermidis* (**A**) and *R. solani* (**B**) in the presence of LysO. 1—5 μg/mL, 2—10 μg/mL, 3—0 μg/mL.

**Table 1 jof-10-00323-t001:** Activities of the extracellular enzymes of *Trichoderma* cf. *aureoviride* Rifai VCM F-4268D in growth media with different carbon sources.

Growth Medium (Carbon Source)	The Enzymatic Activities (units/mL)					
	β-1,3-glucanase	β-1,6-glucanase	Xylanases	Chitinases	Proteases	LysOx
Glucose, 2%	0	0	0	0	0	0
Wheat bran, 3%	14.05 ± 1.2	9.65 ± 0.8	6.10 ± 0.5	2.45 ± 0.2	8.4 ± 0.8	8.2 ± 0.8
Chitosan, 2%	8.5 ± 0.8	4.36 ± 0.3	2.13 ± 0.2	16.45 ± 1.5	5.8 ± 0.5	2.5 ± 0.2
Xylan, 2%	5.2 ± 0.4	3.8 ± 0.3	7.2 ± 0.6	1.4 ± 0.1	4.9 ± 0.4	2.1 ± 0.2
Deactivated mycelium of the *Fusarium decemcellulare*, 3%	11.2 ± 1.1	8.0 ± 0.7	1.8 ± 0.2	12.7 ± 1.1	7.2 ± 0.7	6.8 ± 0.6

## Data Availability

Data are contained within the article.

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
