# Peer review of "Synthesis of Extracellular L-lysine-α-oxidase along with Degrading Enzymes by Trichoderma cf. aureoviride Rifai VKM F-4268D: Role in Biocontrol and Systemic Plant Resistance"

_jof, 2024, doi:10.3390/jof10050323_

Round 1

Reviewer 1 Report

Abstract

The abstract provides a clear overview of the study's objectives and findings. However, it would be beneficial to include specific quantitative results or ranges related to enzyme activities or antimicrobial effects.

How do the enzyme activities and antimicrobial effects vary with different carbon sources or growth conditions?

Introduction

The introduction sets the stage well by highlighting the importance of Trichoderma fungi and LysO in various biological processes. Could you elaborate more on the potential applications of LysO in biotechnology and medicine mentioned in previous studies?

You mention that the synthesis of LysO by Trichoderma fungi was not previously considered in the appropriate relationship with enzyme complexes. Can you provide more context on why this is significant?

Materials and Methods

The methods section provides detailed procedures for enzyme biosynthesis evaluation, antimicrobial activity assays, and ROS measurement. However, it lacks information on the specific strains and media used for culturing test microorganisms. Could you provide more details on this aspect?

How were the samples prepared for mass spectrometric analysis to detect pipecolic acid?

Results

The results section presents comprehensive data on enzyme activities, antimicrobial effects, and ROS generation. It would be helpful to include statistical analyses or significance tests for key findings, especially when comparing different conditions or concentrations.

Can you provide more information on the kinetics of LysO activity and its correlation with the accumulation of hydrogen peroxide and pipecolic acid over time?

Discussion

The discussion elaborates well on the functional role of LysO in Trichoderma fungi and its potential implications in biocontrol and plant resistance. However, it would be beneficial to discuss the limitations of the study or potential confounding factors that might affect the interpretation of results.

Are there any hypotheses or predictions regarding the specific mechanisms by which LysO interacts with microbial cell walls or induces ROS production leading to cell death?

Conclusion

The conclusions summarize the key findings effectively. Could you briefly mention any future directions or follow-up studies suggested by the current findings?

How do the findings of this study contribute to our overall understanding of Trichoderma-fungus interactions and their potential applications in agriculture or biotechnology?

Overall, the manuscript presents valuable insights into the synthesis and functional role of LysO in Trichoderma fungi. Addressing these questions and suggestions would further enhance the clarity, depth, and scientific impact of the article.

Abstract

The abstract provides a clear overview of the study's objectives and findings. However, it would be beneficial to include specific quantitative results or ranges related to enzyme activities or antimicrobial effects.

How do the enzyme activities and antimicrobial effects vary with different carbon sources or growth conditions?

Introduction

The introduction sets the stage well by highlighting the importance of Trichoderma fungi and LysO in various biological processes. Could you elaborate more on the potential applications of LysO in biotechnology and medicine mentioned in previous studies?

You mention that the synthesis of LysO by Trichoderma fungi was not previously considered in the appropriate relationship with enzyme complexes. Can you provide more context on why this is significant?

Materials and Methods

The methods section provides detailed procedures for enzyme biosynthesis evaluation, antimicrobial activity assays, and ROS measurement. However, it lacks information on the specific strains and media used for culturing test microorganisms. Could you provide more details on this aspect?

How were the samples prepared for mass spectrometric analysis to detect pipecolic acid?

Results

The results section presents comprehensive data on enzyme activities, antimicrobial effects, and ROS generation. It would be helpful to include statistical analyses or significance tests for key findings, especially when comparing different conditions or concentrations.

Can you provide more information on the kinetics of LysO activity and its correlation with the accumulation of hydrogen peroxide and pipecolic acid over time?

Discussion

The discussion elaborates well on the functional role of LysO in Trichoderma fungi and its potential implications in biocontrol and plant resistance. However, it would be beneficial to discuss the limitations of the study or potential confounding factors that might affect the interpretation of results.

Are there any hypotheses or predictions regarding the specific mechanisms by which LysO interacts with microbial cell walls or induces ROS production leading to cell death?

Conclusion

The conclusions summarize the key findings effectively. Could you briefly mention any future directions or follow-up studies suggested by the current findings?

How do the findings of this study contribute to our overall understanding of Trichoderma-fungus interactions and their potential applications in agriculture or biotechnology?

Overall, the manuscript presents valuable insights into the synthesis and functional role of LysO in Trichoderma fungi. Addressing these questions and suggestions would further enhance the clarity, depth, and scientific impact of the article.

Author Response

2024-04-19

To reviewer 1

Dear Reviewer:

Thank you so much for your attempts to improve readability of our humble contribution. As about our responses, let me post them right after your comments:

Major comments

Abstract

The abstract provides a clear overview of the study's objectives and findings. However, it would be beneficial to include specific quantitative results or ranges related to enzyme activities or antimicrobial effects.

Edited accordingly

How do the enzyme activities and antimicrobial effects vary with different carbon sources or growth conditions?

The present work did not compare antimicrobial effects depending on carbon sources or growth conditions. Аntimicrobial activity is shown depending on the concentration of LysO and the presence of catalase.

Introduction

The introduction sets the stage well by highlighting the importance of Trichoderma fungi and LysO in various biological processes. Could you elaborate more on the potential applications of LysO in biotechnology and medicine mentioned in previous studies?

Addition information concerning applications of LysO in biotechnology and medicine is included in Introduction.

You mention that the synthesis of LysO by Trichoderma fungi was not previously considered in the appropriate relationship with enzyme complexes. Can you provide more context on why this is significant?

Edited accordingly

Materials and Methods

The methods section provides detailed procedures for enzyme biosynthesis evaluation, antimicrobial activity assays, and ROS measurement. However, it lacks information on the specific strains and media used for culturing test microorganisms. Could you provide more details on this aspect?

Corrected accordingly.

How were the samples prepared for mass spectrometric analysis to detect pipecolic acid?

Corrected accordingly.

Results

The results section presents comprehensive data on enzyme activities, antimicrobial effects, and ROS generation. It would be helpful to include statistical analyses or significance tests for key findings, especially when comparing different conditions or concentrations.

Corrected accordingly.

Can you provide more information on the kinetics of LysO activity and its correlation with the accumulation of hydrogen peroxide and pipecolic acid over time?

We do not yet have the opportunity to measure the accumulation of pipecolinic acid in dynamics and compare it with the accumulation of other metabolites under consideration.   Samples were taken in different experiments and different growth phase in order to show the formation of this compound in general. It was noted, that appearance of pipecolic acid is coincides in time with starting biosynthesis of LysO and the disappearance of lysine.

Discussion

The discussion elaborates well on the functional role of LysO in Trichoderma fungi and its potential implications in biocontrol and plant resistance. However, it would be beneficial to discuss the limitations of the study or potential confounding factors that might affect the interpretation of results.

The main limitation of the study is the impossibility of determining the biomass of the fungus during growth on wheat bran or deactivated fungal mycelium and the evaluation of some physiological parameters, such as specific growth rate, respiratory activity, etc. But we decided do not to cover this issue in the paper.

Are there any hypotheses or predictions regarding the specific mechanisms by which LysO interacts with microbial cell walls or induces ROS production leading to cell death?

Some hypotheses or predictions regarding the specific mechanisms by which LysO interacts with microbial cell walls or induces ROS production are presented currently in Discussion

Conclusion

The conclusions summarize the key findings effectively. Could you briefly mention any future directions or follow-up studies suggested by the current findings?

These results provide new opportunities for regulating fungus-plant-pathogen relationships and obtaining more effective remedies for plant protection and crop storage. The obtained data compliments the strategy of fungus in biocontrol and provides an adaptive advantage in competition with other microorganisms in the natural environment.

How do the findings of this study contribute to our overall understanding of Trichoderma-fungus interactions and their potential applications in agriculture or biotechnology?

Corrected accordingly.

Overall, the manuscript presents valuable insights into the synthesis and functional role of LysO in Trichoderma fungi. Addressing these questions and suggestions would further enhance the clarity, depth, and scientific impact of the article.

Corrected accordingly.

Please let us know if we can do something else to improve the quality of our submission.

Regards

Dr. Alex Vetcher

Reviewer 2 Report

The manuscript reported the biosynthesis of LysO and cell wall degrative enzymes, and revealed the antagonistic effects of LysO on phytopathogenic fungi and bacteria and suggested the potenial factors that determine the antimicrobial activity of LysOs.  Generally, the experimental design is logical and methodology looks reasonable. However, I think a number of questions need to be addressed before it can be accepted.

1. Regarding antimicrobial activity section, a total of nine microorganism strains were mentioned to be tested in the text. However, only three of them were shown in figures (3-5). I wonder if the remaining 6 strains were examined or not. Also, how many replicates were included for each testing? In addition to just showing some picutures, I wonder if calculations of infection area or other relevant index can be performed and added in this section. This may be more convincing than just showing pictures. Further inforamtion needs to be included in figure legend, such as innoculation conditions, scale bar ect. 

2. Figure 1 and Figure 6, I suggest replace line charts with bar charts, and show each enzyme activity in separate graph. Add significant difference. Legend is too simple, please explain every graph in great detail.

3. The introduction section needs improvement, more background info and the significnance of this study should be stated.

4. The reference list is messy, the format is not consistent across references, there are some errors. Also, I think more recent references need to be cited. 

5. A number of language problems were found, including format and grammar errors. For example, H2O2, 2 should be subscript. Please check through the ms carefully. 

The manuscript reported the biosynthesis of LysO and cell wall degrative enzymes, and revealed the antagonistic effects of LysO on phytopathogenic fungi and bacteria and suggested the potenial factors that determine the antimicrobial activity of LysOs.  Generally, the experimental design is logical and methodology looks reasonable. However, I think a number of questions need to be addressed before it can be accepted.

1. Regarding antimicrobial activity section, a total of nine microorganism strains were mentioned to be tested in the text. However, only three of them were shown in figures (3-5). I wonder if the remaining 6 strains were examined or not. Also, how many replicates were included for each testing? In addition to just showing some picutures, I wonder if calculations of infection area or other relevant index can be performed and added in this section. This may be more convincing than just showing pictures. Further inforamtion needs to be included in figure legend, such as innoculation conditions, scale bar ect. 

2. Figure 1 and Figure 6, I suggest replace line charts with bar charts, and show each enzyme activity in separate graph. Add significant difference. Legend is too simple, please explain every graph in great detail.

3. The introduction section needs improvement, more background info and the significnance of this study should be stated.

4. The reference list is messy, the format is not consistent across references, there are some errors. Also, I think more recent references need to be cited. 

5. A number of language problems were found, including format and grammar errors. For example, H2O2, 2 should be subscript. Please check through the ms carefully. 

Author Response

2024-04-19

To reviewer 2

Dear Reviewer:

Thank you so much for your attempts to improve readability of our humble contribution. As about our responses, let me post them right after your comments:

The introduction section needs improvement, more background info and the significnance of this study should be stated.

  1. Regarding antimicrobial activity section, a total of nine microorganism strains were mentioned to be tested in the text. However, only three of them were shown in figures (3-5). I wonder if the remaining 6 strains were examined or not. Also, how many replicates were included for each testing? In addition to just showing some picutures, I wonder if calculations of infection area or other relevant index can be performed and added in this section. This may be more convincing than just showing pictures. Further inforamtion needs to be included in figure legend, such as innoculation conditions, scale bar ect. 

9 strains were used in the studies, indicated in the Materials and Methods section.

Antimicrobial activity and the effect of catalase were carried out for all strains, but only a few are given as examples.

 Intracellular content of ROS in the presence and the absence of LysO have been measured in S. epidermidis and R. solani  (Fig. 6).

      All assays were performed in triplicate including the growth and sterility control.

Inoculation conditions was added to the materials and methods section.

Scale bar added to the figures.

As to infection area or other relevant index - the difference that can be seen in the figure is important here. The absolute size of the lysis zones will not provide additional information to understand the effect of the antimicrobial action.

  1. Figure 1 and Figure 6, I suggest replace line charts with bar charts, and show each enzyme activity in separate graph. Add significant difference. Legend is too simple, please explain every graph in great detail.

In Fig.1 accumulation of extracellulare enzyme activities and H2O2 are presented in dynamics and allow us to compare the temporal and quantitative characteristics of the synthesis of metabolites. Some explanation are presented in the text.

Intracellular content of ROS  is demonstrated as curves in Fig.6 and make it possible to compare the effect of LysO in different concentrations on bacteria and fungi.

  1. The introduction section needs improvement, more background info and the significnance of this study should be stated.

This aspect is presented in the text.

  1. The reference list is messy, the format is not consistent across references, there are some errors. Also, I think more recent references need to be cited. 

Recent references are added to be cited.

  1. A number of language problems were found, including format and grammar errors. For example, H2O2, 2 should be subscript. Please check through the ms carefully. 

corrected accordingly 

 Please let us know if we can do something else to improve the quality of our submission.

Regards

Dr. Alex Vetcher

Reviewer 3 Report

Dear Authors,

I have received your manuscript titled "Synthesis of extracellular L-lysine-α-oxidase along with degrading enzymes by Trichoderma cf. aureoviride Rifai VKM F-4268D: Role in biocontrol and systemic plant resistance", which explores the synthesis of extracellular L-lysine-α-oxidase (LysO) and cell wall degrading enzymes by Trichoderma cf. aureoviride, and its implications for biocontrol and systemic acquired resistance in plants.

After my revision, I acknowledge the significance of your research in contributing to our understanding of Trichoderma cf. aureoviride's role in biocontrol and plant defense mechanisms. The exploration of LysO and its synergistic action with degradative enzymes presents a valuable addition to the field. However, the manuscript, in its current form, faces several topics concerning formatting, cohesion, and depth of analysis that hinder its potential impact and readability.

The introduction, while providing a foundational background, lacks a compelling narrative that situates your study within the broader research context. In the Materials and Methods section, particularly in part 2.4, a detailed explanation of the methodology is essential for reproducibility, and not only a reference. Referencing a previous article is insufficient without a concise description within the manuscript itself.

The Results section's presentation, especially the listing of microorganisms in section 3.1, would benefit from a more structured paragraph format. Furthermore, the absence of standard deviation bars in Figure 1, unlike Figure 6, detracts from the statistical robustness of the presented data. The Discussion section appears fragmented, with short sentences and paragraphs lacking a coherent flow and interconnection. This section would greatly benefit from a more detailed analysis that ties your findings to existing literature and articulates their broader implications more convincingly.

Conclusions need to be more profound, succinctly summarizing the study's contributions to the field and suggesting future research directions. This would provide readers with a clear understanding of the significance and potential impact of your work.

Considering the above, I would recommend a major revision of the manuscript. Specific attention should be paid to improving the manuscript's structure and formatting to adhere to the journal's template, enhancing the narrative flow to ensure clarity and cohesion across sections, and expanding the discussion to offer a more comprehensive analysis of the findings in relation to existing research. Additionally, incorporating detailed methodology within the manuscript and ensuring all figures uniformly present statistical data will significantly improve the manuscript's quality.

Upon thorough revision addressing these concerns, I believe your manuscript could offer valuable insights into the role of Trichoderma cf. aureoviride in biocontrol and plant defense, meriting reconsideration for publication. I look forward to receiving your revised manuscript and appreciate your dedication to enhancing the quality of research in the field.

 Sincerely,

Before consider minor details, the manuscript must be reestructured correctly in all the sections.

Author Response

2024-04-19

To reviewer 3

Dear Reviewer:

Thank you so much for your attempts to improve readability of our humble contribution. As about our responses, let me post them right after your comments:

Microorganisms is not in italics. The manuscript in general is really bad edited and I consider it need a deep revision by authors.

Corrected accordingly

Some subsection are not correctly presented.

Corrected accordingly

Discussion and Conclusion must be reestructured.

Corrected accordingly

The introduction, while providing a foundational background, lacks a compelling narrative that situates your study within the broader research context. In the Materials and Methods section, particularly in part 2.4, a detailed explanation of the methodology is essential for reproducibility, and not only a reference. Referencing a previous article is insufficient without a concise description within the manuscript itself.

Corrected accordingly

The Results section's presentation, especially the listing of microorganisms in section 3.1, would benefit from a more structured paragraph format. Furthermore, the absence of standard deviation bars in Figure 1, unlike Figure 6, detracts from the statistical robustness of the presented data. The Discussion section appears fragmented, with short sentences and paragraphs lacking a coherent flow and interconnection. This section would greatly benefit from a more detailed analysis that ties your findings to existing literature and articulates their broader implications more convincingly.

Corrected accordingly

Conclusions need to be more profound, succinctly summarizing the study's contributions to the field and suggesting future research directions. This would provide readers with a clear understanding of the significance and potential impact of your work.

Corrected accordingly

Considering the above, I would recommend a major revision of the manuscript. Specific attention should be paid to improving the manuscript's structure and formatting to adhere to the journal's template, enhancing the narrative flow to ensure clarity and cohesion across sections, and expanding the discussion to offer a more comprehensive analysis of the findings in relation to existing research. Additionally, incorporating detailed methodology within the manuscript and ensuring all figures uniformly present statistical data will significantly improve the manuscript's quality.

Corrected accordingly

Please let us know if we can do something else to improve the quality of our submission.

Regards

Dr. Alex Vetcher

Round 2

Reviewer 1 Report

I have reviewed the revised version of the manuscript, and I would like to acknowledge the authors' efforts in addressing the comments and suggestions provided during the initial review process. The revisions made have substantially improved the clarity, depth, and scientific impact of the manuscript.

The authors have appropriately addressed all major and minor concerns raised in my previous review. They have provided comprehensive responses to each comment, and the revisions made reflect a thorough consideration of the feedback provided.

I have reviewed the revised version of the manuscript, and I would like to acknowledge the authors' efforts in addressing the comments and suggestions provided during the initial review process. The revisions made have substantially improved the clarity, depth, and scientific impact of the manuscript.

The authors have appropriately addressed all major and minor concerns raised in my previous review. They have provided comprehensive responses to each comment, and the revisions made reflect a thorough consideration of the feedback provided.